# A Multi-Agent-Based Defense System Design for Multiple Unmanned Surface Vehicles

Shangyan Zhang [1], Weizhi Ran [1], Geng Liu [2], Yang Li [2] and Yang Xu[1,*]

1. School of Computer Science and Engineering, University of Electronic Science and Technology of China, Chengdu 611731, China
2. System Engineering Research Institute, China State Shipbuilding Corporation Limited, Beijing 100094, China
* Correspondence: xuyang@uestc.edu.cn; Tel.: +86-189-8094-7876

**Abstract:** Defense systems are usually deployed to protect high-value targets or hot spots that are integral parts of the modern battlefield environment. However, in coastal defense operations (due to the variability of the maritime environment and the sustainability of combat), limited operational capabilities, the need for efficient coordination, and protracted combat are peculiarly challenging to meet by traditional manned fleets. In contrast, with lower costs, unmanned fleets can organize an autonomous defense against enemy targets that are capable of rapid response. This paper focuses on the typical defense scenario; we analyzed and modeled the objective functions of the intelligent defense system and propose a hierarchical distributed multi-agent-based system design scheme. Finally, to test the system's performance, we established simulation verification experiments in a typical scenario and compared the system based on the traditional central architecture. The results show that, in a defense operation, the hierarchically-distributed multi-agent-based system shows improvements in system decision-making efficiency and interception effect.

**Keywords:** multi-agent system; USVs coordination; hierarchical distributed system; defense scenario

## 1. Introduction

In a typical defense operation environment, defense systems need to be deployed near high-value targets or disputed areas to form warning areas with specific defense ranges. Defense operations using small, unmanned vessels are increasingly used in coastal areas and regions difficult to access by large vessels [1–3]. Multiple units cooperate to form a defense force to intercept targets that have invaded the warning area [4–6]. In the coastal defense field, in recent years, due to the low costs and high mobility of USVs [7,8], more USVs have been equipped [9–11]. Multiple USVs compose intelligent systems with high performance and reliability through efficient cooperation, which significantly improves the operational efficiency of the defense system.

For multi-agent-based defense systems, the system architecture often determines the dominant relationships among agents. In recent ten years, the multi-agent-based system design has attracted widespread attention through its application in various typical scenarios. In the research of multi-agent-based systems, the prominent structures are divided into two types [12–14], one is the thoroughly centralized system, and the other is the thoroughly distributed system.

The advantage of a thoroughly centralized system is the globally centralized decision-maker. Usually, by constructing an algorithm with low complexity, a globally optimal scheme can be generated according to the global state. Reference [15] provides a dynamic load balancing algorithm, but it is merely applied to small-scale systems. Reference [16] provides a centralized algorithm based on emotional attenuation, which can produce the single task assignment plan of a group of heterogeneous robots.

On the other hand, the thoroughly distributed system is different (not planned by a central agent). Each agent can perceive the environment, exchange information with

other surrounding agents, and complete independent planning. Every agent can share the computing pressure of each decision loop in the system. Therefore, it has good adaptability in a large-scale dynamic environment. Based on the auction process, Reference [17] involves an incremental task allocation algorithm based on the contract net protocol. Still, it cannot meet the dynamic needs of the system.

In the actual application of sea defense, due to the following dynamic uncertainties, the thoroughly central and distributed systems cannot improve the efficiency of defense combat [18].

- Dynamic uncontrollable factors of the environment;
- The number of enemies detected is uncertain;
- The ability of the enemy is uncertain;
- Defense forces available are constantly changing during each decision cycle;
- Defense forces are heterogeneous.

When facing these dynamic environmental factors, the thoroughly central system needs to ensure uninterrupted information interaction to help the central node complete each decision-making process, which tends to bring about a 'central node overload' under a heavy load. In the decision-making process of a typical fully distributed system, due to the changes of uncertain factors [19], the information exchange among all agents will produce a large amount of communication burden, which has certain requirements for the communication bandwidth of the system.

This paper focuses on how the system utilizes USVs to form a scheme to intercept multiple enemy targets after the warning area detection in the coastal defense operation. Under the objective function of the defense system, then based on the characteristics of the central system and distributed system, considering the cooperative actions among agents, a hierarchically-distributed multi-agent-based defense system design scheme is proposed. The central task allocation between the center node and groups improves decision-making efficiency, while distributed collaboration among the agents within the group enhances the dynamic characteristics of the system. Finally, a typical coastal defense scenario was designed to experiment. Compared with the centralized system and static grouping agents, the results show that in a defense operation the hierarchically-distributed multi-agent-based system shows improvements in system decision-making efficiency and interception effect.

The rest of this paper is presented as follows. In Section 2, two cooperation modes among agents in interception tasks are analyzed. Moreover, In Section 3, the objective function of the defense system is generated. Then, a hierarchical distributed multi-agent-based defense system design is presented in section 4. In Section 5, to test the system, a defense simulation in a typical scenario is run and all results of the experiment are presented. Finally, we present our concluding remarks by summarizing our contribution and experimental results in Section 6.

## 2. Cooperation Modes

To solve the interception task, the system needs to build multiple agents with different functions to form a defense power. Moreover, agents must form and maintain the best formation that can be complete in the current state and meet the following requirements as much as possible:

1. The formation configuration should form a convex polygon (as far as possible) and surround the target inside the polygon. There must be obstacles in the forward direction of the target.
2. When the target moves, the formation should be able to adjust the position of the formation in real-time according to the moving speed of the target, and always put the target inside the formation.
3. The formation members are replaceable, and can complete the corresponding independent planning according to the target state.

For a single agent, the behavior process of the agent is generally divided into three steps: observation, decision, and action, which are consistent with the OODA loop theory of unmanned systems [20,21]. However, due to the complex dynamics of the environment, the simple addition of multiple OODA loops cannot accomplish complex tasks [22]. The goals can be achieved only by generating joint action with a cooperative relationship among agents. The cooperative relationship of agents is mainly reflected in the intersection of OODA loop processes. In this problem, there are two coordination types among agents, one is hierarchical coordination and the other is decentralized coordination.

Hierarchical multi-agent-based coordination mainly includes the action sequence in decision-making or the relationship between superiors and subordinates. Generally, before making the next decision, an agent's decision-making process needs to wait for the result of its superior or other agents' decision. Moreover, an agent can trigger the decision process only after receiving the command from its superior agent, as shown in Figure 1.

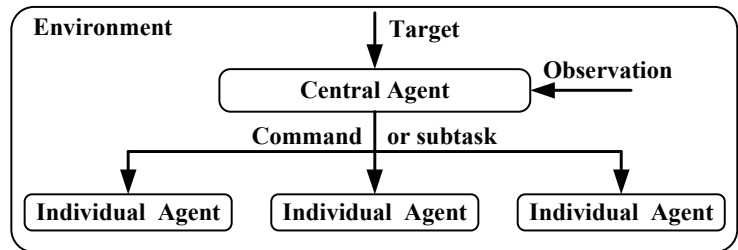

**Figure 1.** Hierarchical multi-agent-based coordination in interception tasks.

In the defense system, it can be found that there is a central agent in the defense area, and a fleet formed by multiple individual agents. In the process of formation, the central agent assigns main tasks to other agents in different groups. After the formation process is completed, the group agents complete the subtask allocation according to the group task goal, so that each agent can obtain the individual goal. This process is tallied with hierarchical multi-agent-based coordination.

Decentralized coordination exists in the case that agents must take similar joint actions to achieve the same goal. As shown in Figure 2, the interaction of collaborative information is often bidirectional. Even under certain circumstances, the positions or roles of agents in collaboration can be interchanged with each other.

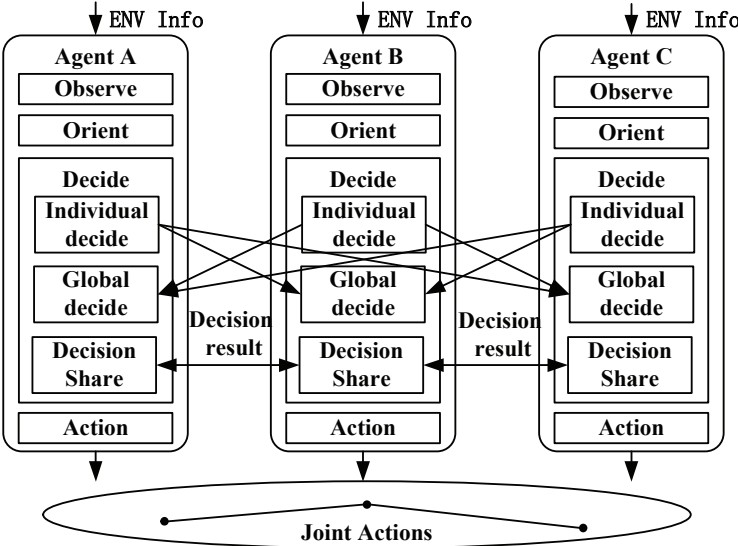

**Figure 2.** Decentralized multi-agent-based coordination in interception tasks.

Therefore, after the formation is made up, decentralized coordination is formed among individual agents in each group. Firstly, in the observation stage of each USV [23], the agents not only use their perceptron to observe the external environment but also exchange state information among agents. After obtaining the state information of all members in the group, agents choose their best position in the formation and re-exchange information and then make action decisions independently to form joint actions with other members.

## 3. Defense Objective Function Modeling

According to the analysis of cooperation mode in Section 2, to construct a multi-agent-based system, it is necessary to model the defense objective function. Formally, at a certain time, the multi-agent-based system $A_m$, $A_m = \{a_1, a_2 \ldots, a_m\}$ detects many targets $T$, $T_n = \{t_1, t_2 \ldots, t_n\}$. To intercept enemy targets, the system divides all agents into different groups $G_n$, $G_n = \{G_1, G_2, G_3 \ldots G_n\}$, and each group $G_k$ has a corresponding target $t_j$.

To accomplish the interception task, agents in a group must form and maintain the best formation so that the fleet always puts the target inside the formation. Thus, the individual capability $C$ difference between agent and target, such as speed, steering, and endurance, will lead to system resource consumption. The cost function of the $t_j$ interception by $a_i$, such as in Equation (1), is composed of two major parts, one comes from the distance and the other from the capability gap.

$$cost(a_i, t_j) = L(P_{a_i} - P_{t_j}) + H(C_i^a, C_j^t) \tag{1}$$

1. $L$ is the energy consumption sailing from position vector $P_{a_i}$ to $P_{t_j}$;
2. $H$ is the power cost between individual capabilities $C_i^a$ and $C_j^t$.

Here, the objective is to survive the intrusion with minimal loss and cost of resource utilization by maximizing targets being intercepted while minimizing the cost of operations to be able to defend against subsequent engagements. If a group with a size of $N_{G_k}$ intercepts the set target $t_j$ successfully, agents in the group can obtain the reward $V_j$. So, the utility of group $G_k$ is calculated using Equation (2).

$$U_{kj} = N_{G_k} V_j - \sum_{a_i \in G_k} Cost(a_i, t_j) \tag{2}$$

Hence, the multi-agent system's objective can be formulated, such as in Equation (3).

$$F = Max(\sum_{G_k \in G_n, t_j \in T_n} U_k j) \tag{3}$$

To enable all agents to intercept the target efficiently, algorithm 1 is used to help the multi-agent-based system to generate a grouping scheme quickly and the groups can be dynamically adjusted based on observation information. To define how many USVs are needed to stop a ship and prevent all USVs from joining one group, which would lead to the defense power concentrated, the algorithm needs to set the maximum size of the group. However, in general, all USVS can be well organized by utility $U_{kj}$ computing.

## 4. Multi-Agent-Based Defense System Design

For the thoroughly centralized system, as shown in Figure 3a, the agents in this system cannot form decentralized coordination. During interception operations, the individual decision-making needs to be finished by the central node [24]. All information should be summarized to the central node, so it is difficult to deal with the dynamism of objectives under the tremendous computation [25]. For the thoroughly distributed system, as shown in Figure 3b, the agents in this system cannot form hierarchical coordination, and the formation scheme cannot be quickly formed through negotiation by distributed agents. Under the global redundancy information produced by all individual agents, it is too tough to handle the huge overall interception task.

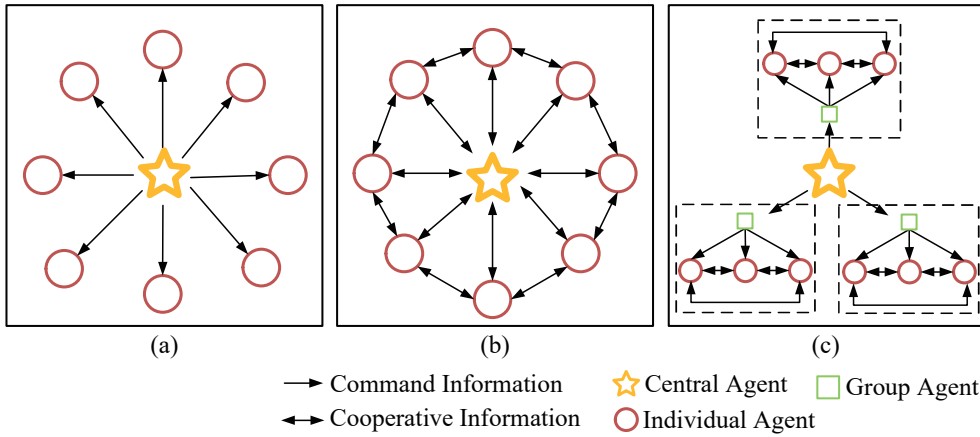

→ Command Information  ☆ Central Agent  ☐ Group Agent
↔ Cooperative Information  ○ Individual Agent

**Figure 3.** Typical multi-agent-based defense system architectures. (**a**) The thoroughly central system. (**b**) The thoroughly distributed system. (**c**) The hierarchical distributed system.

---

**Algorithm 1:** Dynamic grouping algorithm based on the defense objective function

---

**Input:** AgentList *Agents*, GroupList *Groups*,TargetList *Targets*, the maximum size of group *max_size*

**Output:** *Groups*

1 **for** $i = 1; i <= |Targets|; i + +$ **do**
2     Observe state of $t_i$
3     $P_{t_i} \leftarrow t_i's$ position,$C_i^T \leftarrow t_i's$ ability
4     **for** $j = 1; j <= |Agents|; j + +$ **do**
5        $P_{a_j} \leftarrow a_j's$ position, $C_j^A \leftarrow a_j's$ ability
6        $cost(a_j, t_i) = L(P_{a_j} - P_{t_i}) + H(C_j^A, C_i^T)$
7        Update agent's *cost* of $t_i$
8     **end**
9     Sort *Agents* by *cost*
10     Initial group $G_k$
11     **for** $k = 1; k <= |Agents|; j + +$ **do**
12        Obtain $a_k$ and join $G_k$
13        $U_{ki} = N_{Gk}V_i - \sum_{a_k \in G_k} Cost(a_k, t_i)$
14        **if** $U_{ki} > 0$ **then**
15           break
16        **end**
17        **if** $G_k's$ size>=*max_size* **then**
18           break
19        **end**
20     **end**
21     Issue $t_i$ to $G_k$
22     *Groups* add $G_k$
23 **end**
24 **return** *Groups*

---

Combined with the characteristics of the centralized and distributed systems, this paper proposes a hierarchically-distributed system architecture, as shown in Figure 3c. An overall interception task can be divided into multiple simple subtasks by a central agent, and a group composed of several distributed agents can achieve subtasks. Therefore, this hierarchical distributed system architecture can meet the two cooperation modes among agents in Section 2. In addition, all groups can obtain local observations and share meaningful information with the central agent. With the central agent and several groups, all agents can finish their own decision process quickly under smooth information circulation.

Based on the hierarchical distributed architecture and the defense objective function in Section 3, a multiple USV cooperative defense intelligent system, as shown in Figure 4, is mainly composed of three types of agents, central decision-making agent, group collaborative agent, and individual agent.

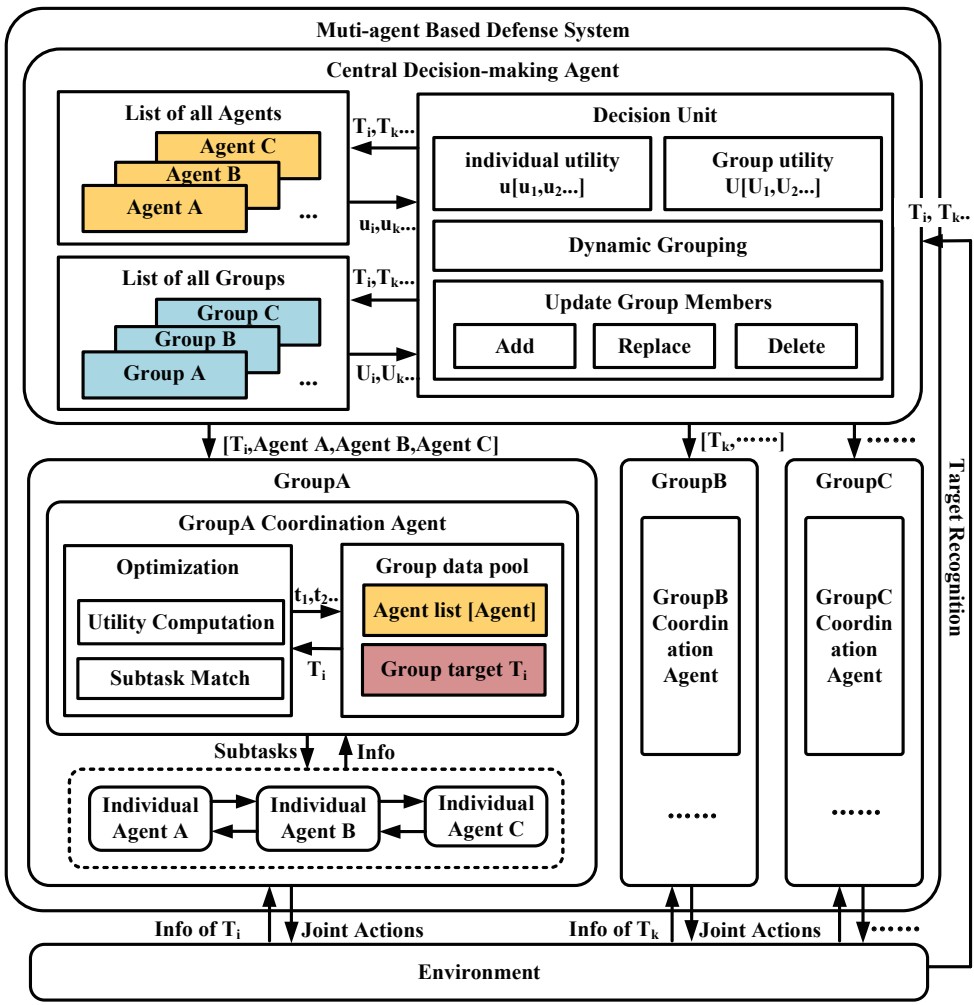

**Figure 4.** The multi-agent-based defense system design and data streams.

The central decision-making agent is mounted on the shore Command Centre or large boats that can use high-power detectors to detect enemy targets in the defense area and build a target list according to the result of the target recognition function. Moreover, the central decision-making agent has a global list of individual agents and collaborative agents. Through the calculation of individual utility and group utility, the dynamic grouping algorithm is applied to generate and update the group, and continuously optimize the composition of each group according to the current target information. Then, the central decision-making agent issues the current target information to each group.

After receiving the subtask from the central decision-making agent, the group cooperative agent mainly cooperates with other individual agents within the group and subdivides the group objectives according to the utility model of each agent. At the same time, the group cooperative agent can obtain the local state of other agents, calculate the target utility, and add or remove members according to the currently observed target information and group state information.

The individual agent is the main component of each group. After receiving the individual target, it interacts with other agents in the group and shares observation information to plan individual behaviors that form consensus and joint actions. In addition, in the process of interception, individual agents can adapt to the dynamics of the environment

by observing the state of the targets. All agents can also make independent decisions according to their utility model, to a certain extent.

## 5. Experiments and Results

To test the system capability, a typical multiple USV cooperative defense scenario was established. In this scenario, the system was constructed based on the hierarchically-distributed system design and many dynamic groups were generated to intercept the intruding enemy ships. At the same time, the control group based on the centralized system uses the static grouping method to intercept targets. The experiment set-up for this work is described as follows.

In the given coastal area, as shown in Figure 5, there were some protected targets at the coast. Two parties were involved in the sea area around the protected targets: the combat units in the defending interior area (referred to as the ally faction) and the periphery units (herein referred to as the enemy faction). Our central ship and several USVs formed a base area with a specific warning red boundary, and a defense area with a specific green boundary was formed at the periphery. Enemy ships entered from the edge of the defense area and invaded our protection targets along a particular track. The defense system should dispatch USVs to intercept these enemy ships.

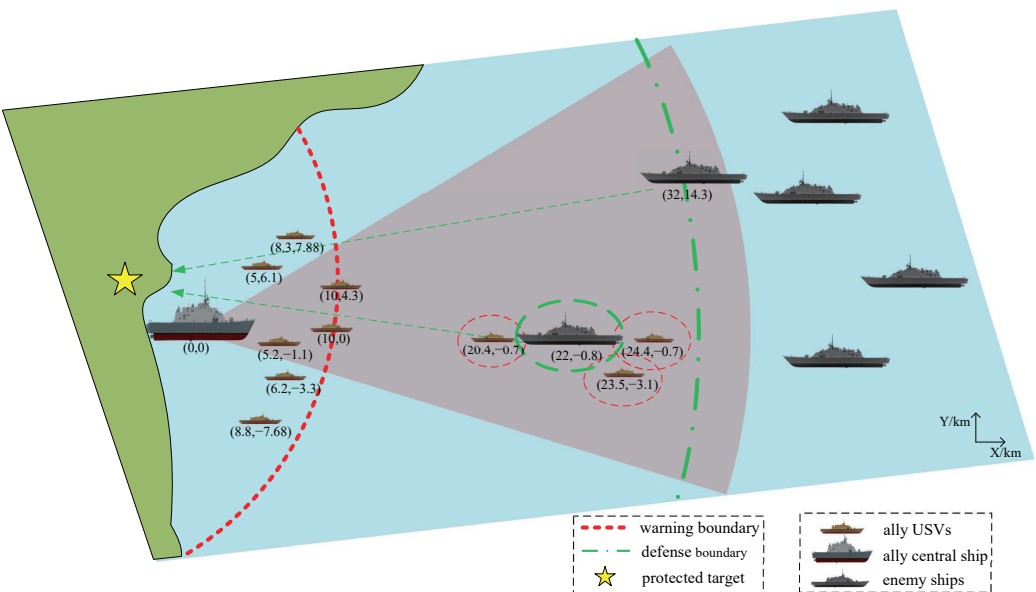

**Figure 5.** A typical coastal defense scenario with multi-USV cooperation.

The ally faction consisted of 1 center ship and 10 combat USVs, and the number of enemy ships was variable. A coordinate system was established based on the center ship as the origin (0,0), and other ally combat USVS were randomly distributed around the center ship. USVs and enemy ships have the following capabilities, as shown in Table 1. At the beginning of each experiment, each capability value of the ship is set as a random value in the range.

**Table 1.** Capability settings in the defense scenario.

| - | Speed/kn | Turning Diameter/m | Durability |
|---|---|---|---|
| Ally USVs | [10, 20] | [1, 3] | [200, 500] |
| Enemy ships | [5, 10] | [4, 5] | - |

Here, the maximum size of the static group is set to 3, indicating that the interception group is composed of 3 unmanned surface vessels. The enemy ship will decelerate under the interception of the formation composed of USVs until the speed is 0, and then the enemy

ship will return to the start point. Figure 6 shows a typical successful defense scenario. In the intrusion process, once an enemy ship enters the red boundary of the base area, the defense mission of this group fails. Similarly, if an enemy ship leaves the defense area after the interception operation, the invasion of the ship fails.

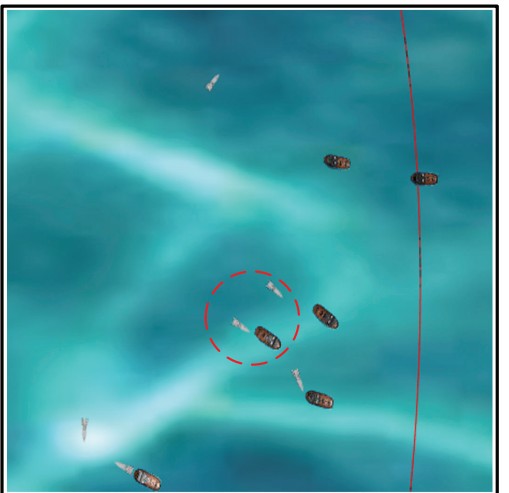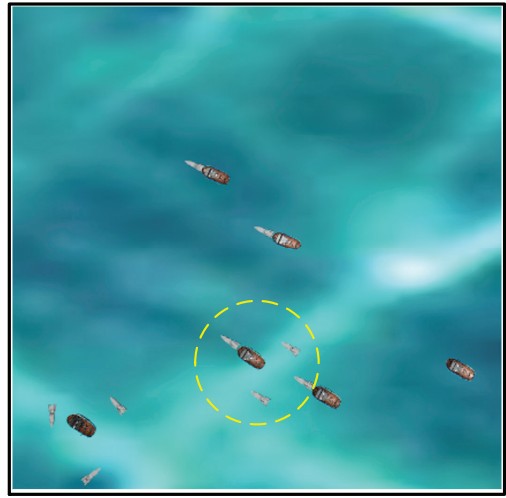

**Figure 6.** Typical confrontation details in the scenario.

In addition, various random parameters were initialized at the beginning of each experiment. With the same parameters, comparative experiments were carried out via dynamic grouping and static grouping systems, respectively. Firstly, the number of enemy ships was set to 1. Afterward, we increased the total number of enemy ships continuously in the replicate experiment. The defense system performance of the experiment was analyzed based on three performance indexes as follows.

- Efficiency of the defense system according to the invasion distance of the enemy.

The enemy's invasion distance refers to the depth of the enemy's invasion, namely, the total sailing distance of all enemy ships after they cross the defense boundary. The lower the invasion distance, the better the interception effect of the system. When the total number of enemy ships increased from 1 to 25, we calculated this index during each experiment, as shown in Figure 7a; the total enemy intrusion distance of the dynamic groups was always lower than that of the static groups. Accordingly, the interception effect of the hierarchical distributed system based on the dynamic grouping method is better than that of the centralized system.

- Efficiency of the defense system according to the number of the invaded enemy.

Furthermore, we repeated the experiment by increasing the number of enemy ships from 1 to 40 and counting the number of enemy ships that invaded the base boundary. The lower the number, the better the interception effect of the system. As shown in Figure 7b, when the number of enemy ships increased to 20, the static grouping system had considerable interception pressure. Until the total number of enemy ships increases to 30, the dynamic grouping system can still intercept enemy ships effectively.

- Efficiency of the defense system according to the duration of interception.

The duration of interception refers to the time from when the operation starts to when it ends, including the decision time. In addition, due to the fact that some failed interceptions will advance the end time, it is necessary to add some penalty time. Hence, the total interception time of the multi-agent system can be formulated as Equation (4). The lower the duration of interception, the more efficient the execution of the system.

$$time_{interception} = time_{decision} + time_{operation} + time_{penalty} \qquad (4)$$

When the interception system is not effective, the simulation will end soon, which will lend to the illusion of a short interception time. To correctly test the system, the penalty time is necessary when some enemy ships invade successfully. Here, $time_{penalty}$ is the product of $N_f$ and $k$. Wherein, $N_f$ is the number of successfully invaded ships, and $k$ is the penalty coefficient (set to 1000). We repeated the experiment by increasing the total number of enemy ships from 1 to 30 and then computed the duration of interception.

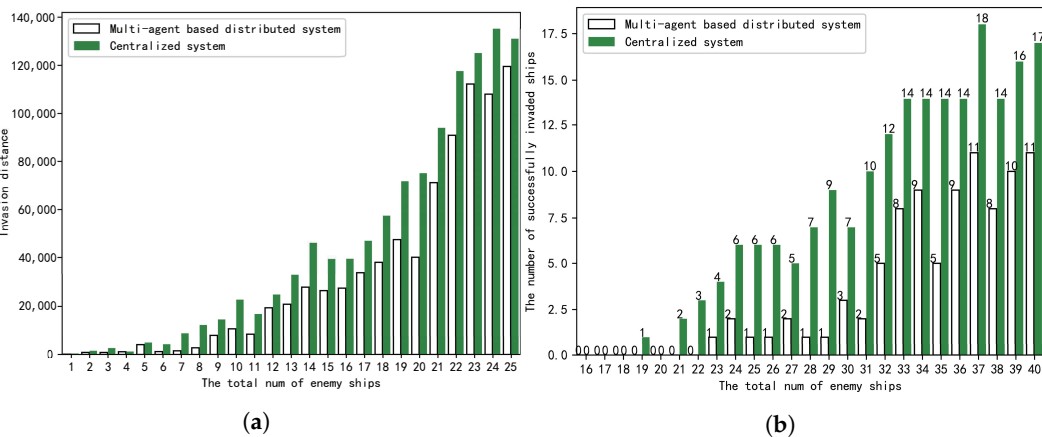

| (a) | (b) |
|---|---|

**Figure 7.** Experimental results on the invasion performances of the enemy ships. (**a**) The total invasion distance of all enemy ships. (**b**) The total number of successfully invaded ships.

As shown in Figure 8, it can be found that when the total number of enemy ships is small, the static group barely spends more time. When the total number of enemy ships is greater than 20, in each experiment, the duration of interception of the hierarchically-distributed system is much less than the centralized system, indicating that when the number of enemies is small, the dynamic grouping system has a shorter reacting time, faster decision-making, and more efficient execution.

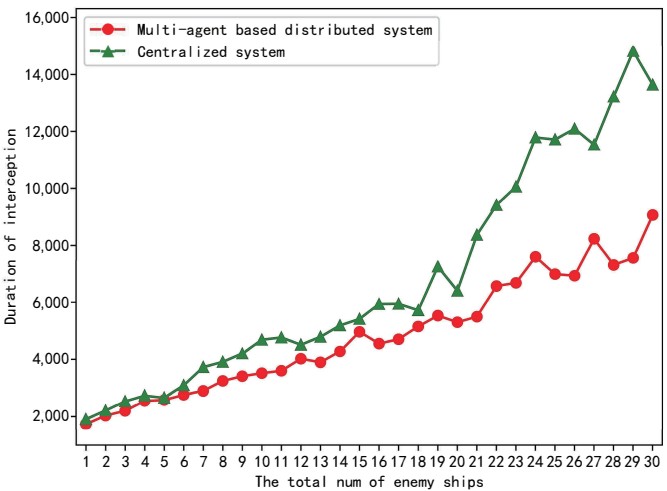

**Figure 8.** Experimental results about the duration of interception.

## 6. Conclusions

This paper constructs a hierarchically-distributed multi-agent-based system (combined with the characteristics of the central and distributed systems) to solve a USV defense problem in the field of coastal defense. In addition, the experimental verification was carried out in the typical cooperative defense scenario, and a centralized system was set as the control group. The results show that with the increase in the total number of enemy ships, the hierarchically-distributed multi-agent-based system has a better interception

effect. Performance indexes, such as the invasion distance, the number of successfully invaded ships, and the duration of interception, show that the multi-agent-based system designed in this paper has an exceptional dynamic defense ability, can be applied to coastal defense, and protect important areas or high-value targets.

**Author Contributions:** Conceptualization, Supervision and funding acquisition, Y.X.; methodology, software, validation, writing—original draft preparation, S.Z.; software and validation, writing—review and editing W.R.; software, investigation and validation, writing-review and editing, G.L.; software and validation, investigation, Y.L. All authors have read and agreed to the published version of the manuscript.

**Funding:** This research received no external funding.

**Conflicts of Interest:** The authors declare no conflict of interest.

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
