# Peer review of "A Multi-Agent-Based Defense System Design for Multiple Unmanned Surface Vehicles"

_electronics, doi:10.3390/electronics11172797_

Round 1
Reviewer 1 Report
Authors present a novel methodology to dispatch USVs in groups to protect an inland target. USVs command is based on a hierarchical distribution. Their work is based on the OODA loop including a collaborative and hierarchical layer.
The article is well written and organized. After introducing the proposed architecture and algorithm, they present a software simulation to prove their pertinence.
I found the article interesting and well written. Nevertheless, I think some details and justifications are needed to better understand your work.
In algorithm 1, I could not find the maximum size of a group that you describe in line 197. This maximum size should be discussed because it is important to define how many USVs are needed to stop a ship.
I am curious about time interception (Equation 4). Can you give some examples of time? I feel that time decision is insignificant. In this equation, I think it is better for the reader to say “time_penalty” and explain and justify this variable afterwards. I am not sure if I well understood why the penalty.
I could not find any information in the article about your testing software. I think it is important to share some information to know if it is possible to easily test other strategies. It is open source?
A lot of people print articles in black and white (me included). It was not easy to make the difference between allies and enemies in Figure 5. Also, the style of the lines of the boundaries is the same, so it is difficult to identify them. I found the same problem in Figure 7 (red and green looks similar in black and white).
Author Response
We thank the respected reviewer for his suggestion. I send the response in the attachment.

Reviewer 2 Report
I send the review in the attachment.

Author Response
Thanks so much for your valuable comments. I send the response in the attachment.
